# Social Identity Dimensions as Drivers of Consumer Engagement in Social Media Sports Club

**Željka Marčinko Trkulja [1], Jasmina Dlačić [1] and Dinko Primorac [2,\*]**

[1]   Faculty of Economics and Business, University of Rijeka, 51000 Rijeka, Croatia
[2]   Department of Business Economics, University North, 42000 Varaždin, Croatia
\*   Correspondence: dinko.primorac@unin.hr

**Abstract:** Consumer engagement is defined as a multidimensional concept in this study via the identification of members of a sports club social network and consumer identification with a sports club brand on social networks. Hence, the study focuses on sports clubs that engage with their customers through social media. The purpose of this paper is to provide a conceptual and theoretical understanding of consumer engagement, particularly among sports teams that employ interactive platforms to establish relationships with customers. Furthermore, in order to better comprehend the interaction between consumer and brand, this research approaches customer identification as consisted of two distinct constructs. Consumer identification with members of a sports club social network is separated from customer identification with a sports club brand. Research results point out that the consumer identification with members of the sports club's social network and consumer identification with the sports club's brand are positively related to consumer engagement. Furthermore, the value creation process is enhanced if the sport club approaches separately different dimensions of customer engagement in the social networks of the sports club.

**Keywords:** consumer identification; consumer engagement; social networks; sport clubs





## 1. Introduction

Research in the field of digital marketing management in the sports industry is particularly relevant today, given the evident growing trend in the application of digital transformation of the sports brand. With the increase in the acceptance of the Internet and social networks, in terms of a new communication channel, its strategic importance in the marketing of the sports industry is also growing. Research in the field of digital marketing indicates that marketing strategies must be clearly identified in order to maintain and ensure the competitive advantage of companies in the digital environment (Kannan and Li 2017). A sustainable competitive advantage can only be achieved by combining the company's existing capabilities and using digital technologies. The sports industry recognizes the usefulness and importance of strategic marketing planning. It has been found that without the implementation of strategic planning in marketing, sport will not survive in a competitive entertainment industry market (Kriemadis and Theakou 2007). As the business environment becomes more complex and consumer needs become more sophisticated, the need for companies, especially in the sport industry, to carefully analyze and implement strategic planning in marketing is evident.

Use of digital technology in this process helps companies in the sport industry to achieve more adequate market positioning.

On social networks, consumers have the ability to filter, select, and share information with each other or with the company (Nambisan and Baron 2007). The result of this participatory digital medium is the transformation of consumers into both recipients and creators of content. As such social media influencers (SMIs) gain their importance over influencing consumer's decision-making processes (Cheung et al. 2022), the strength of

consumer influence on the attitudes and opinions of other members of the virtual community is commensurate with the influence of the brand's marketing message and its cultural significance (de Valck et al. 2009). This social variable influences consumer purchasing decisions. Consumers are subject to social influence and consider the perceptions and judgments of other individuals relevant (Demiray and Burnaz 2019). It is evident that susceptibility to social influence stems from the tendency to acquire product knowledge by seeking information from others, adapting to others' expectations to gain reward or avoid punishment, and identifying with others through buying and owning certain products or brands (Demiray and Burnaz 2019). Moreover, influencer-promoted ads outperform in customer engagement and positive sentiment from followers compared to the identical ads placed by brands (Lou et al. 2019).

As customer engagement on the sport club's social network is still under researched phenomenon. Due to different forms of engagement that emerge among customers, and as their identification with both other members as well as brand is identified as crucial in this process. Hence, the purpose of this paper is to explore consumer identification with the virtual (social network) community of a sports club brand and its impact on consumer engagement in social media sports club. Hence, this paper builds on social identity theory (Mael and Tetrick 1992), customer engagement theory (Pansari and Kumar 2017), and customer engagement in social media (de Oliveira Santini et al. 2020). Thus, we aim to contribute to a better understanding of customer engagement in social media in the sport industry and to provide insight on how customer identification contributes to customer engagement especially when dealing with sports clubs' social networks.

This paper is structured as follows. In Section 2, a literature review and theoretical background on customer engagement and social identity and its dimensions is provided. In Section 3, the research methodology and results are presented. In Section 4, a discussion is offered, and theoretical and practical contributions are identified. In Section 5, the paper ends with identified limitations and further research ideas.

## 2. Literature Review

### 2.1. Consumer Engagement

Consumer engagement on social networks is manifested through an active relationship with the brand given that the interaction between consumers and businesses through social networks allows consumers to play an active role during online experiences (Santos et al. 2019). Consumer engagement on social networks can be seen as a result of the consumer's experience with the brand, through the process of value co-creation encouraged by the company and the relationship between consumers of the virtual community of the brand, product or service (Santos et al. 2019). Understanding and adapting the concept of consumer engagement within the sports industry can achieve a variety of goals, including (Filo et al. 2015): increase market share, increase sales, reduce costs, improve brand perception, increase consumer confidence, increase consumer satisfaction and consumer loyalty. Consumers eager to participate in sports and spectator sports competitions as well as lovers of healthy lifestyles, influenced the transformation of the sports industry. Hence, the sport industry has become the fastest growing sector whose daily growth is noticeable in all sport areas, including entertainment, fitness, as well as professional and academic sports (Filo et al. 2015).

Sports clubs reward consumer engagement on the social media. Social networks are focused on strengthening and enhancing the participatory culture of sports club virtual brand community members and a sense of belonging using a system of preferences, retweets, sharing, personalization, and upgrades (Parganas et al. 2015). Social networks help to create a sense of community by encouraging members of the sports club's virtual community to participate and interact (Filo et al. 2015). It was the advent of the internet and social networks that solved the insufficiently interactive relationship between sports consumers and clubs, which was most often caused by expensive tickets for the match, geographical restrictions or the inability to watch the match on television due to broadcasting

problems. These barriers have been overcome by enabling communication between sports clubs and consumers, live streaming, video and audio, and timely posting on social media (Parganas et al. 2015).

Despite the significant contribution of previous research to the understanding of marketing factors, such as the quality of social networks, information shared in the digital environment or stimuli that direct the consumer to continuously visit the social network of a sports club (Pronschinske et al. 2012; Sashi 2012) there is limited understanding and empirical research on how to conceptualize and measure consumer engagement with a sports club on social media (Filo et al. 2015).

Two concepts have been proposed for understanding consumer engagement with sports clubs (Yoshida et al. 2014). The first concept refers to consumer behavior in the role of a fan (interest in attending a sports event, watching and reading news about a sports club), and consumer behavior outside the role of a fan (behavior directed towards the sports club and other fans based on the moral values of the fan). The second concept analyzes consumer activity related to transactional behavior (e.g., repeat purchase, media consumption, consumption) and non-transactional behavior (e.g., interaction with other fans and sports club). Accordingly, the consumer's engagement with the sports club can be defined as the consumer's spontaneous, interactive, and co-creative behavior with the sports club and/or other consumers in order to achieve individual and social benefit (Yoshida et al. 2014). However, as Yoshida et al. (2014) did not focus on digital environment, which represents an important platform for interaction with sports club fans (Filo et al. 2015), there is still a gap in the research to cover.

Sports consumers engage by attending sporting events of their favorite teams, watching broadcast matches, buying licensed club merchandise, reading and discussing news in sports periodicals and newspapers, and talking about sports with other fans and engaged consumers (Hunt et al. 1999; Bristow and Sebastian 2001; Funk and James 2001). A customer that is engaged in sports is more likely to direct their attention not only on activities that benefit him personally (e.g., attending sporting events, following sports through media, reading, and shopping) (de Ruyter and Wetzels 2000; Swanson et al. 2003), but also focus on activities that benefit their favorite sports clubs, e.g., showing off his fandom, giving good reviews on social media or in web portals, attending sporting events, as well as activities with other fans, e.g., sharing knowledge about the club, communicating with each other, being supportive, and encouraging in the fan atmosphere).

In this study, consumer engagement is approached as a multifaceted concept that includes cognitive, affective and behavioral components of engagement. These three components are further developed into sub-dimensions of consumer engagement, such as attention, absorption, acceptance, enthusiasm, enjoyment, learning, and sharing, (Dessart et al. 2015). Consumer engagement is expressed in a variety of behaviors that result in a better consumer relationship with the sports club, which extends typical consumer loyalty indicators such as frequency of visits, buying behavior, and future buying intentions (Vale and Fernandes 2018).

### 2.2. Social Identity and Its Dimensions

Understanding the identification process is an important input to help strategic decision-making and contribute to the optimal allocation of efforts to promote consumer identification with a brand, a virtual community, or both. Analysis of various aspects of identification will allow to more adequately encourage consumers to spread a positive recommendation and direct consumer activities useful to businesses and their products (Gwinner and Swanson 2003).

According to Tajfel and Turner (1986), social identity is defined as the part of an individual's self-concept that results from the awareness of being a member of a particular group, but also from the emotional significance of belonging to that group. Because consumers are often associated with a number of brands and identification attributes, (Ambler et al. 2002), analyzing only one factor does not lead to an adequate understanding of the characteristics of the brand identification concept. Consumers can make decisions through information

from sources with high credibility and trust and Influencers' opinions will matter more for their Followers' decisions on engaging with the brand (Argyris et al. 2020). Therefore, it is necessary to consider both consumer identification with the virtual community of the sports club brand and consumer identification with other consumers of the virtual community of the sports club. In this sense a variety of brand consumer relationships contribute to the interaction (Muniz and O'Guinn 2001).

Analysis of social communities, including the possibility that the virtual community influences the perceptions and activities of other members (Schau et al. 2009) is important. Consumer interaction in social communities is extensive and affects the speed of information dissemination (Kozinets et al. 2010), influences perceptions of consumer reviews of new products and offerings, and these activities increase the possibility of engagement and collaboration with loyal consumers (Franke and Shah 2003). Personal identification has been found to influence consumer engagement with a sports club brand (Algesheimer et al. 2005).

The consumer identification process within the brand communities of the sports club often starts with establishing the brand consumer relationship. The consumer can self-categorize by identifying the brand, establish relationship with it, and observe others c sharing the same enthusiasm for the brand (Algesheimer et al. 2005). Thus, the relationship with members of the sports club brand community influences the relationship with a sports club brand. Consequently, personal identification with the social community of the sports brand may result in an increase in consumer engagement on the social networks of the sports brand.

Two types of identification in brand social networks exists (Dessart et al. 2015): with members of the virtual community and with the brand The theory of social identification is mainly applied to consumer identification with the brand, in which variables such as brand prestige, level of expectations, length of membership and frequency of contact with the brand were studied (Popp et al. 2016). The aforementioned variables are also applied to this research on consumer identification with the brand of a sports club in order to examine the role of identification. The degree to which the brand expresses and enhances the consumer's identity is determined by the level of brand identification. User experiences on brand social communities direct participants to interact with the virtual community, which is personified by other individuals within the group, as well as to interact with the brand, which is moderated by the group's corporate administrator (Wirtz et al. 2013). Extending the conceptualization from Wirtz et al. (2013), it is evident that engagement on a brand's social network is triggered by numerous factors arising from relationships with the brand, social values of the community, as well as functional aspects of online membership in the brand's virtual community (Dessart et al. 2015). A consumer's relationship with a chosen brand appears to foster consumer engagement in the community (Dessart et al. 2015). By associating with the brand and what the brand stands for, members of virtual brand communities feel a sense of intimacy with other community members (Algesheimer et al. 2005).

Brand identification of consumers has psychological advantages, such as strengthening consumer self-esteem (Wann and Branscombe 1990) and results in consumer loyalty, product purchases and electronic word-of-mouth (E-WOM) (Ambler et al. 2002), willingness to pay a higher price, and resilience to negative company information (Rubio and Marin 2015). The relationship with the brand and consumer interaction on consumer social networks has been identified as a factor influencing consumer engagement (Brodie et al. 2013). A virtual brand community cannot exist without the consumer identifying with the brand (Muniz and O'Guinn 2001).

The concept of consumer identification with a brand is a sense of connection between the consumer and the company and represents the extent to which consumers perceive unity with the company (Ross et al. 2008). Identification is therefore consensually defined as the psychological state of consumer perception, consumer feelings, and evaluation of consumer belonging to a company (Rubio and Marin 2015).

Social networks embody amounts of data about users and lend themselves very well to analytics, as this data is published by the users themselves and is closely linked to their attitudes, beliefs, interests and personality in general. In recent years, an increasing number of analyses use data from social networks (Erevelles et al. 2016). Hence, there is a need to examine consumer identification with sports club social network members and other concepts related to the sports clubs social networks to understand the nature and characteristics of consumer engagement in such communities. Despite the fact that participation in communities where members share similar preferences is an important factor in brand success (Algesheimer et al. 2005; Vale and Fernandes 2018), consumer engagement with other members of a sports club's virtual brand community has not been extensively researched in the context of spectator sports. Previous research focuses on engagement with product or service brands or companies (Bowden 2009; van Doorn et al. 2010). The focus of this research is brand engagement (Gummerus et al. 2012; Vivek et al. 2012; Hollebeek and Chen 2014; Wallace et al. 2014). However, engagement with other market stakeholders, such as other consumers, can also greatly influence a brand or company (Brodie et al. 2011) as recognized in the literature on brand communities (Algesheimer et al. 2005; Schau et al. 2009).

## 3. Conceptual Model and Hypotheses Development

Despite the prevalence of digital media, there is a distinct dearth of studies on consumer engagement on social networks of sports clubs. Consumer engagement on digital platforms of sports club brands may increase confidence in the sports club brand, as well as stimulate the growth of value co-creation and the sports club brand, by increasing consumer satisfaction and loyalty. Consequently, consumer engagement has caused a change in the application of brand management in the sports industry (Vale and Fernandes 2018). For sports clubs, understanding and applying consumer engagement enables them to realize a strategic advantage because consumer engagement stimulates consumers to use the sports club's offer more actively. An engaged sports consumer will actively look for the opportunity to attend specially organized sports events of the sports club he supports, such as attending matches, and is especially motivated to attend events that include socializing with the sport's club players.

Consumer engagement is reflected in a variety of behaviors that lead to stronger consumer loyalty to the sports club, beyond traditional metrics of consumer loyalty, such as frequency of visits, buying behavior, and future consumer intentions (Vale and Fernandes 2018). The affective dimension of engagement encompasses consumer emotion towards the engagement focus (Calder et al. 2017) and follows pattern of continuous emotional process.

The need to examine various aspects of consumer identification with the virtual community of a sports club brand and its impact on consumer activities and behavior in the sports industry is evident. Given the importance of consumer identification with the virtual community of the sports club brand, the following hypothesis was set:

**Hypothesis 1 (H1).** *Consumer identification with members of the sports club's social network has a positive impact on consumer engagement on the sports club's social networks.*

Consumer identification with a brand has been found to be a key determinant of brand loyalty (Zaglia 2013), but it also influences determinants of behavior in the digital environment, such as consumer engagement through social media and positive e-WOM (Chu and Kim 2011). Given the importance of identifying consumers with the brand of a sports club, the following hypothesis was set:

**Hypothesis 2 (H2).** *Consumer identification with the sports club brand on social networks has a positive impact on consumer engagement on the sports club's social networks.*

According to literature review and stated hypotheses, the following conceptual model is proposed (Figure 1).

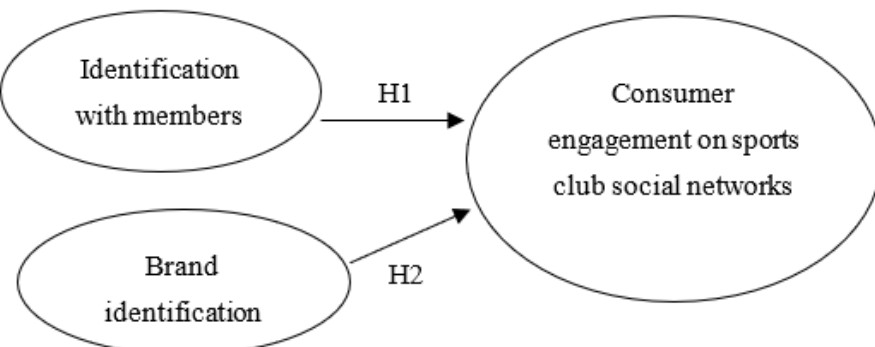

**Figure 1.** Conceptual model.

## 4. Research Methods

### 4.1. Measurement

Measures included in the research are consumer identification with the brand, consumer identification with members of a sports club's social network and consumer engagement. All scales were adapted to the context of the spectator sports and used 7-point Likert scale anchored at "1—I completely disagree" to "7—I completely agree". To avoid the response bias, scale items were dispersed throughout the questionnaire.

Consumer engagement was measured on a scale from Dessart et al. (2016). According to this scale, consumer engagement consists of the emotional, behavioral, and cognitive dimensions divided into 7 sub-dimensions. The emotional dimension integrates: enthusiasm and enjoyment. The cognitive dimension encompasses: attention and elements of absorption. While the behavioral dimension includes: sharing, learning, and acceptance.

Consumer identification with members of a sports club's social network was measured on a scale from Algesheimer et al. (2005). Research dimensions are identification with the virtual community, engagement within the virtual community, intention to continue membership, intention to recommend the virtual community, and behavior within the virtual community. The consumer identification with the brand was measured on the scale of Popp et al. (2016).

### 4.2. Data Collection and Sample Profile

A survey of consumers and fans of sports clubs in the field of spectators' sports, namely basketball, football and handball were conducted. These group spectators' sports were selected based on the popularity of sports clubs and the number of followers on social networks. Data were collected through an online questionnaire via www.1ka.si (accessed on 1 June 2019). The research sample included 322 respondents (research participants) who follow sports clubs in the field of basketball, football and handball on social networks. Considering the sample size of 322, which makes the sample a large sample, all samples above 30 respondents (Field 2009) where, according to the central limit theorem, the data behave as if they were normally distributed, no additional data checks will be performed regarding their normality of distribution.

The analysis of the results used descriptive statistics for analyzing the socio-demographic characteristics and describing the research results, bivariate statistics and multivariate statistics for analyzing the research hypotheses. IBM SPSS version 20 (IBM Corp. Released 2011. IBM SPSS Statistics for Windows, Version 20.0. Armonk, NY, USA: IBM Corp.) and R statistical programs (R Core Team (2021). R: A language and environment for statistical computing. R Foundation for Statistical Computing, Vienna, Austria. URL https://www.R-project.org/) were used for statistical analysis.

Research sample consists of 52% of male and 48% of female respondents. The largest share of respondents, 67%, have a residence in the City of Zagreb. The analysis of the age structure showed that half of the respondents (50%) belong to the youngest age group, i.e.,

they are between 18 and 24 years old. At the same time, 63% of respondents are less than 35 years old, while 37% of respondents are more than 34 years old.

## 5. Research Results

Table 1 presents descriptive statistics of the consumer identification with social network members sports club and consumer identification with the brand of the sports club's social network items.

**Table 1.** Selected descriptive statistical indicators of individual identification indicators.

|  |  | Valid Answers | Average Value | SD |
|---|---|---|---|---|
| Consumer identification with social network members sports club | I share the same goals with the members of the sports club's social community/social network that I follow on social media. | 322 | 3.920 | 2.052 |
|  | I see myself as part of a sports community virtual community/social network community. | 321 | 3.860 | 2.043 |
|  | I am connected to the members of the social network of the sports club. | 322 | 3.670 | 2.117 |
|  | Friendship with members of the sports club's social network is significant to me. | 322 | 3.520 | 2.093 |
|  | If members of a sports club's virtual community/social network planned something, I think it would be something we would "do" rather than something they would "do" if we planned it together. * | 321 | 2.920 | 1.861 |
|  | My actions are often influenced by the expectations of the members of the sports club's social network. | 322 | 2.450 | 1.853 |
| Consumer identification with the brand of the sports club's social network | The image of a sports club social network and my image are congruent in many respects. * | 322 | 3.110 | 1.945 |
|  | I consider myself a valuable member of the social network of the sports club I follow. * | 321 | 3.110 | 2.012 |
|  | When someone praises the social network of the sports club I follow, I understand it as a personal compliment. | 320 | 2.840 | 2.010 |
|  | I would experience an emotional loss if I had to stop following the social networks of the sports club. | 322 | 2.410 | 1.907 |
|  | I believe others respect me for my connection to the sports club's social network. | 322 | 2.320 | 1.717 |

Note: * item omitted after exploratory factor analysis. Source: Research results.

The analysis of individual dimensions of identifications determined that the highest value belongs to the item related to the notion that they share the same goals with members of the virtual community/social network of the sports club they follow on social networks (M = 3.920). Meanwhile, the lowest value belongs to the item related to the notion that the believe others respect them because their connections with the social network of the sports club (M = 2.320). The average value of items in the scale Consumer identification with social network members sports club is M = 3.39, while the scale Consumer identification with the brand of the sports club's social network is M = 2.758.

Subsequently, to continue the analysis exploratory factor analysis was performed. After checking the adequacy of data using the KMO measure and the Bartlett test, it

was decided that analysis could continue using the principal axis factoring method with Oblimin rotation.

In order to maintain both theoretically predicted constructs of identification, the a priori criterion was used as a method for determining the factors to be retained, i.e., a predetermined number of desired factors was given, which is two. In an attempt to detect the most adequate factor structure, items with a saturation of less than 0.3 were omitted from further analysis, according to Field (2009). During the analysis three items were omitted. The final factor structure of the individual's identification presented two factors, i.e., Consumer identification with social network members sports club and Consumer identification with the brand of the sports club's social network, that explain 62.072% of the total variance of the instrument.

Table 2 shows the matrix of the factor form of the final factor solution.

**Table 2.** Result of exploratory factor analysis of individual's identification.

| Items | Factor Saturations | |
|---|---|---|
| | Identification with Members | Brand Identification |
| I am connected to the members of the social network of the sports club. | 0.892 | |
| I share the same goals with the members of the virtual community/social network of the sports club I follow on social media. | 0.784 | |
| Friendship with members of the sports club's social network is significant to me. | 0.732 | |
| I see myself as part of a sports community virtual community/social network community. | 0.593 | |
| I believe others respect me for my connection to the sports club's social network. | | 0.851 |
| I would experience an emotional loss if I had to stop following the social networks of the sports club. | | 0.808 |
| When someone praises the social network of the sports club I follow, I take it as a personal compliment. | | 0.645 |
| My actions are often influenced by the expectations of the members of the sports club's social network. | | 0.529 |
| Eigenvalue | 4.142 | 0.823 |
| Percent of the variance explained | 51.781% | 10.291% |
| Cronbach's Alpha | 0.868 | 0.844 |

Source: Research results.

The instrument adopted is reliable as the Cronbach's Alpha reliability coefficient is for both factors above the threshold of 0.7 (Nunnally 1978), that is 0.868 for Consumer identification with social network members sports club and 0.844 for factor Consumer identification with the brand of the sports club's social network. Hence, analysis can continue using the identified factors.

The measuring instrument was intended to measure consumer engagement on social networks of sports clubs consisted of seven theoretically predicted dimensions of consumer engagement (Enthusiasm, Enjoyment, Attention, Absorption, Sharing, Learning, Approval). Selected descriptive statistical indicators of individual indicators are shown in Table 3.

**Table 3.** Selected descriptive statistical indicators of consumer engagement on social networks of sports clubs items.

| | | Valid Answers | Mean Value | Standard Deviation |
|---|---|---|---|---|
| Enthusiasm | I am delighted with the sports club I follow on social media. | 320 | 5.780 | 1.368 |
| | I am interested in everything related to the sports club that I follow on social networks. | 322 | 5.300 | 1.689 |
| | The sports club I follow on social media is interesting. | 321 | 5.240 | 1.716 |
| Endorsing | I say positive things about the sports club to other people/persons. * | 321 | 5.210 | 1.772 |
| | I promote/actively support a sports club on a social network. | 322 | 4.660 | 2.079 |
| | I try to get other people interested in the sports club I support. | 321 | 3.870 | 2.202 |
| | I actively defend the sports club on the social network against its critics. * | 321 | 3.320 | 2.193 |
| Enjoyment | Socializing and interacting with a sports club on social media makes me happy. | 320 | 4.200 | 1.952 |
| | I feel energetic in contact with the sports club I follow on social media. | 322 | 3.670 | 1.973 |
| | Interacting with a sports club on social media is like a reward to me. | 322 | 2.930 | 1.927 |
| Attention | I spend a lot of time thinking about the sports club I follow on social media. | 322 | 3.390 | 2.092 |
| | I always find time to think about a sports club. | 321 | 3.020 | 2.031 |
| Sharing | I share my ideas on the social networks of the sports club. | 322 | 2.860 | 2.04 |
| | I share interesting facts on the social networks of the sports club. | 320 | 2.850 | 1.915 |
| | I help the sports club on social media. | 320 | 2.840 | 2.071 |
| Learning | I ask questions on the social networks of the sports club. | 321 | 4.390 | 2.025 |
| | I'm looking for ideas or information on the sports club's social network. * | 322 | 2.640 | 1.965 |
| | I'm asking for help on the sports club's social network. * | 322 | 2.510 | 1.811 |
| Absorption | I forget about everything around me when I interact with a sports club on social media. | 320 | 2.930 | 2.047 |
| | Time flies when I interact with a sports club on social media. | 321 | 2.710 | 1.87 |
| | I get carried away when I interact with a sports club on social media. | 322 | 2.660 | 1.942 |
| | I am reluctant to interrupt my interaction with the sports club on social networks. | 321 | 2.430 | 1.871 |

Note: * item omitted after exploratory factor analysis. Source: Research results.

Considering the dimensions of consumer engagement on social networks of sports clubs, the highest average value is noted for dimension Enthusiasm (M = 5.44), while the lowest average value belongs to the items in dimension Absorption (M = 2.68).

To continue the analysis among dimensions of consumer engagement on the social networks of sports clubs, exploratory factor analysis was performed. After checking the adequacy of data using the KMO measure and the Bartlett test, it was decided that analysis can continue using the principal axis factoring method with Oblimin rotation. In order to maintain theoretically predicted dimensions of consumer engagement on the social networks of sports clubs, an a priori criterion was used as a method for determining the factors that should be retained, i.e., a predetermined number of desired factors was specified. As the exploratory factor analysis identified six factors, opposed to theoretically seven, the factor Learning saturated by only one item was omitted from further analysis. According to the suggestion by Field (2009) items with a saturation of less than 0.3 were omitted from further analysis. The final factor structure of the consumer engagement on the social networks of sports clubs explains 73.922% of the total variance of the instrument and contains factors Enjoyment, Enthusiasm, Sharing, Attention, Endorsing, and Absorption. Table 4 shows the matrix of the factor pattern of the final factor solution.

**Table 4.** The result of the exploratory factor analysis of consumer engagement on the social networks of sports clubs.

| Items | Factor Saturations | | | | | |
| --- | --- | --- | --- | --- | --- | --- |
| | Enjoyment | Enthusiasm | Sharing | Attention | Endorsing | Absorption |
| I feel energetic in contact with the sports club I follow on social networks. | 0.679 | | | | | |
| Interaction with a sports club on social networks is like a reward for me. | 0.516 | | | | | |
| Socializing and interacting with the sports club on social media makes me happy. | 0.349 | | | | | |
| The sports club I follow on social media is interesting. | | 0.816 | | | | |
| I am delighted with the sports club I follow on social networks. | | 0.774 | | | | |
| I am interested in everything related to the sports club I follow on social networks. | | 0.715 | | | | |
| I share interesting things on the sports club's social networks. | | | 0.872 | | | |
| I share my ideas on the sports club's social networks. | | | 0.865 | | | |
| I ask questions on the sports club's social networks. | | | 0.822 | | | |
| I help a sports club on social media. | | | 0.683 | | | |
| I spend a lot of time thinking about the sports club I follow on social media. | | | | 0.797 | | |
| I always find time to think about the sports club. | | | | 0.702 | | |
| I try to get other people interested in the sports club I support. | | | | | −0.877 | |

**Table 4.** *Cont.*

| | Factor Saturations | | | | | |
|---|---|---|---|---|---|---|
| **Items** | **Enjoyment** | **Enthusiasm** | **Sharing** | **Attention** | **Endorsing** | **Absorption** |
| I promote/actively support a sports club on a social network. | | | | | −0.528 | |
| Time flies when I interact with a sports club on social media. | | | | | | 0.873 |
| I get carried away when I interact with a sports club on social media. | | | | | | 0.865 |
| I forget about everything around me when I'm interacting with a sports club on social media. | | | | | | 0.743 |
| I am reluctant to interrupt the interaction with the sports club on social networks. | | | | | | 0.683 |
| Eigenvalue | 9.101 | 1.621 | 1.312 | 0.516 | 0.390 | 0.366 |
| Percent of the variance explained | 50.562% | 9.003% | 7.291% | 2.866% | 2.165% | 2.035% |
| Cronbach's Alpha | 0.878 | 0.855 | 0.906 | 0.864 | 0.797 | 0.929 |

Source: Research results.

The instrument adopted is reliable as the Cronbach's Alpha reliability coefficient is for all identified factors are above the threshold of 0.7 (Nunnally 1978) and ranging from 0.797 for Endorsing to 0.929 for Absorption.

Based on the results of explorative factor analysis for consumer identification with social network members sports club, consumer identification with the brand of the sports club's social network, and dimensions of consumer engagement on the social networks of sports clubs (enjoyment, enthusiasm, sharing, attention, endorsing and absorption), new factors were formed as the average score of included items. These factors were used for testing hypotheses. The method used for hypotheses testing is multiple regression analysis with Enter as the selection method. Several multiple regression analyses were performed with dimensions of consumer engagement on the social networks of sports clubs (enjoyment, enthusiasm, sharing, attention, endorsing and absorption) as dependent variables. Consumer identification with social network members sports club and consumer identification with the brand of the sports club's social network were in all multiple regressions used as independent variables. Results are presented in Table 5.

**Table 5.** Multiple regression results.

| | Dimensions of Consumer Engagement on Social Networks of Sports Clubs | | | | | |
|---|---|---|---|---|---|---|
| **Dimensions of Identification** | **Enjoyment** | **Enthusiasm** | **Sharing** | **Attention** | **Endorsing** | **Absorption** |
| Identification with members | 0.308 | 0.373 | 0.408 | 0.143 | 0.451 | 0.097 * |
| Brand identification | 0.537 | 0.105 | 0.382 | 0.558 | 0.254 | 0.666 |
| $R^2$ | 0.586 | 0.198 | 0.504 | 0.430 | 0,409 | 0.532 |
| $R^2$ adj | 0.583 | 0.193 | 0.501 | 0.427 | 0.405 | 0.529 |
| F-value | 222.188 | 38.590 | 158.652 | 118.919 | 109.036 | 177.882 |

Note: All data are statistically significant at $p < 0.001$ except for * $p < 0.05$.

Based on performed multiple regression analysis, we can conclude that both consumer identification with social network members sports club and consumer identification with the brand of the sports club's social network do influence consumer engagement on the social networks of sports clubs with different magnitude; with the highest influence of

consumer identification with social network members sports club on Endorsing (β = 0.451), and the highest influence of consumer identification with the brand of the sports club's social network on Absorption (β = 0.666). However, to compare their influences on consumer engagement on the social networks of sports clubs, we calculated the average scale for all dimensions of consumer engagement on the social networks of sports clubs. Results are presented in Table 6.

**Table 6.** Hypothesis testing.

| Hypothesis | Hypothesis Testing | The Decision on Hypothesis |
|---|---|---|
| H1. Consumer identification with members of the sports club's social network has a positive impact on consumer engagement on the sports club's social networks | B = 0.373 ** | The hypothesis is accepted |
| H2. Consumer identification with a sports club brand on social networks has a positive impact on consumer engagement on the sports club's social networks. | B = 0.528 ** | The hypothesis is accepted |

Note: ** $p < 0.001$.

Based on research analysis, we can conclude that both hypotheses (H1 and H2) were accepted. Consumer identification with a sports club brand on social media has a higher influence on consumer engagement on a sports club's social networks (β = 0.528) than Consumer identification with members of a sports club's social network (β = 0.373).

## 6. Discussion and Conclusions

This research established and tested a model in which the drivers of consumer engagement on the social network of a sports club, such as individual's identification with members of sports club's social network and with sports club brand, influence consumer engagement on the sports club's social network. The relevance of the topic under study is seen in the understanding and adaptation of the consumer engagement concept within the spectators' sports sector. Consumer engagement in sports club social networks is reflected in various behaviors that lead to a stronger relationship between consumers and a sports club that goes beyond traditional measures of consumer loyalty, such as attendance frequency, purchase behavior, and future intentions (Vale and Fernandes 2018), increasing brand perception, consumer trust, or consumer satisfaction. Hence, focusing on consumer engagement will contribute to the sport club.

This paper aimed at contributing to a better understanding of customer engagement in social media in sport industry and to provide insight on how customer identification contributes to customer engagement especially when dealing with sports clubs' social networks. Paper contributes in several ways. First, it demonstrates that individual's identification both with the members or brand on sports' club social media consequently influences their consumer engagement. This research explored the significance of distinguishing consumer identification with members of a sports club social network and with a sports club brand. This is in line with previous research on online and offline identification (Algesheimer et al. 2005; Popp et al. 2016) indicating the importance of approaching customer's identification as multifaceted construct. Additionally, by differentiating between various identification goals, it is possible to positively influence the value co-creation (Prahalad and Ramaswamy 2004; Marčinko Trkulja 2021). According to this study, customer engagement on sports club social networks is heavily driven by brand identity. It is obvious that having distinct identification aims is critical for optimizing the value generation process in the digital world.

Second, we approach consumer engagement on a sports club's social networks as multidimensional structure of six different dimensions, thus offering the possibility to understand consumer engagement according to different dimensions when spectators' sports on social media are explored.

Third, approaching consumer engagement on a sports club's social networks as a multidimensional construct offers the idea that individual's identification with members of a sports club's social network or with a sports club brand on social media influences differently each customer engagement dimension. As indicated, individual's identification influence dimensions of customer engagement on a sports club's social networks differently. This provides the possibility to sport marketing managers to identify some of the engagement dimensions as especially important for their sport club, to encourage such behavior. For example, if a specific sport club wants to encourage enthusiasm among its spectators, it is advised to boost the identification among social network members to feel the spirt of connectedness and mutual sharing through communication on the social media.

Customer identification on social networks is approached in the framework of social-identity theory. This helped to understand the complex relationships between consumers and sports club brand. Moreover, this research contributes to the understanding of customer engagement in social media as a multidimensional construct when sport club's social media is considered.

In addition, implications for marketing managers of sport clubs are noted. The results can be used by marketing managers to understand customer engagement on social network of a sports club as a multifaceted. Hence, marketing managers by identifying a dimension of customer engagement that is important to them to boost, they can approach their sport customers or fans by encouraging such engagement through posts on social media and communication narrative. Hence, if identification with members is found to be more influencing specific dimension, then in posts, reels, or stories, marketing managers can stress friendship among members through posts from matches, socializing during the match or before/after the match, to boost identification with members and consequently provoke sharing dimension of consumer engagement. Similar, if identification with brand provides greater support to the selected dimension of consumer engagement, then marketing managers can encourage brand identification through a sense of respect if someone is following a sport club social network, e.g., posting how it is important to follow the social media to keep up with the latest news and prioritizing social media over other communication channels, thus providing the sense of absorption of sport customers and fans in information they provide, such as additional information on players, news from the locker room, or behind the scenes footage after the match.

Since the research used the convenient sample, generalizing research conclusions to all sports clubs that use digital media for communicating with target audience is limited. Still, as the research sample size is rather big, it could be used as good indication of relations in the spectator sports. Moreover, a more detailed analysis of the consumer engagement on the social networks of the sports club is needed. This could help grasp in more depth the phenomenon, so additional relationships and possible identifications with not just brand or social network members can be analyzed. Through qualitative research, such as in-depth interviews and focus groups with members active on social networks of sport clubs, researchers could obtain some additional insights in regard to the richness of the relationships observed. This research is limited to one country. Hence, using a multi country sample would contribute to the generalizability of research results. Further research could also include differentiating the sample by sports or by frequency of interaction with other members through social media networks of the sports club.

**Author Contributions:** Conceptualization, Ž.M.T. and J.D.; methodology, Ž.M.T.; software, Ž.M.T., and J.D.; validation, Ž.M.T., J.D. and D.P.; formal analysis, Ž.M.T.; investigation, J.D. and D.P.; resources, Ž.M.T. and J.D.; data curation, Ž.M.T., J.D. and D.P.; writing—original draft preparation, Ž.M.T. and D.P.; writing—review and editing, J.D. and Ž.M.T.; visualization, Ž.M.T. and D.P.; supervision, J.D.; project administration, Ž.M.T.; funding acquisition, D.P. and J.D. All authors have read and agreed to the published version of the manuscript.

**Funding:** This research was funded by University of Rijeka grant number [ZIP-UNIRI-130-8-20].

**Data Availability Statement:** The data presented in this study are available on request from the corresponding author.

**Conflicts of Interest:** The authors declare no conflict of interest.

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
