# Peer review of "Social Identity Dimensions as Drivers of Consumer Engagement in Social Media Sports Club"

_jrfm, doi:10.3390/jrfm15100458_

Round 1

Reviewer 1 Report (Previous Reviewer 3)

The paper clearly identifies its implications for practice and for further research. The implications presented are consistent with the findings and conclusions of the paper. In addition, the new version of the paper adequately addressed the observations and suggestions made by the reviewers in the other evaluation rounds.

Author Response

The authors are grateful for Reviewer 1 comments. 

Reviewer 2 Report (Previous Reviewer 4)

The paper provides some interesting findings. I recommend the editor to accept the paper. 

Author Response

The authors are grateful for the Reviewer 2 comments. 

This manuscript is a resubmission of an earlier submission. The following is a list of the peer review reports and author responses from that submission.

Round 1

Reviewer 1 Report

This paper tries to identify some antecedents of the consumer engagement on a sports' club social media.  Data has been collected for this purpose, but the paper as it is now, it is not ready to be published for more essential reasons:

  • there is not a literature review section to illustrate how the concepts are presented in the extant literature. References to the literature related to the subject are included in the materials and methods section (that usually has another purpose). However, the references included are limited and the literature is not up-to-date (the most recent bibliographical reference - just one- is dated in 2018). The theoretical background in not appropriately presented.
  • the methodology is not fully explained and information is not presented in a logical order. It is not clear how, where from, when the data was collected.
  • the results section should better clarify the methods used for data analysis.
  • there is no discussion section in the paper. The authors do not relate their own findings with the existing literature.
  • the theoretical and practical contributions of the paper need to be presented.
  • the abstract of the paper does not present what it is in the paper.
  • the paper looks as if it was extracted from another material and it is no finalized for publication: many editing mistakes, sentences not finished, the numbering of tables/references to them incorrect, some comments in the paper look as if they do not belong (ex: the Learning sub-dimension). It looks like the authors send a draft.
  • English needs serious proofreading. Some sentences could not be understood (partly English, partly unfinished/unclear content).
  • the journal's format requirements are not respected.

Author Response

Thank you for this observation and comment. 

We restructured the paper: added more elaborated literature review section, rewritten materials an methods section and added new up-to date references. Hopefully this adds to the more adequate presentation of the theoretical background as well as more focused materials in methods section.

We have rewritten this section to be more logically structured and added missing explanations about data collection.

We amended this section and stated the methods used for the data analysis. 

We added Discussion section and elaborated out findings in relation to other researchers. 

We added theoretical and practical contributions in the Conclusion section. 

We have rewritten the abstract. And hope it is now more narrowly pointing out what is in the paper. 

We have rewritten the paper and sent it to the English proof-reader. 

We applied journal guidelines on structuring the paper.

Reviewer 2 Report

Although, the paper is interesting but authors must elaborate the gap in the research work how it differs than the latest study in this area and also to mention in the Abstract and Introduction.

The motivation and contribution of this study are not clear.

Therefore, the innovation and novelty of the article should be emphasized in they.

Please enhance the review of related literature by highlighting the gaps in literature and emphasizing the novel aspects of your work.

In the article isn’t the discussion. 

The conclusion section has to expand the discussion on the policy implications of the empirical findings.

Author Response

Thank you for your observation!

We added theoretical and practical contributions in the Conclusion section. 

We have rewritten the abstract. And hope it is now more narrowly pointing out what is in the paper.

We added theoretical and practical contributions in the Conclusion section. 

Reviewer 3 Report

Ref. jrfm-1690153

Title: SOCIAL IDENTITY DIMENSIONS AS DRIVERS OF CONSUMER ENGAGEMENT IN SOCIAL MEDIA SPORTS CLUB

Journal: Journal of Risk and Financial Management

Thank you for the opportunity to review the paper titled: “SOCIAL IDENTITY DIMENSIONS AS DRIVERS OF CONSUMER ENGAGEMENT IN SOCIAL MEDIA SPORTS CLUB”. The author explored the consumer engagement and social networks of sports clubs. This is a timely issue and interesting. However, the manuscript has several critical problems.

1) Please specify the strategy of strategic sports marketing planning.

2). p. 2 line 50-51. Does a brighter society influence consumer decisions? In this context, do you mean sports consumers?

3). This is an interesting paper and, in large part, I found it enjoyable to read. However, like the other reviewer, I found substantial holes in the paper overall, which will require a thorough re-write and re-conceptualization.

   In the main, areas to develop fall under:

  1. a) The introduction must be expanded and restructured in order to highlight the conceptual delimitation mentioned in the article. We recommend highlighting the novelty of the study according to the evidence of previous studies.
  2. b) There is a lack of importance literature review which makes it difficult to evaluate where this research sits within the field.

4). It is recommended that the hypothetical inferences be written separately from the methodology to give a clearer context.

5). Could the authors talk more as to the reliability and validity of the measures used? Why these measures?

6). The paper clearly identifies its implications for practice and for further research. The implications presented are consistent with the findings and conclusions of the paper.

Author Response

(The authors gave the same response as above.)

Reviewer 4 Report

This study provides some interesting findings for both academics and managers. However, there are several shortcomings and the authors are recommended to amend the paper according to the following suggestions: 

  1. The importance of engagement should be justified by using recent statistics (from statista) , especially for the sports clubs. 
  2. Some recent findings about engagement are overlooked, the authors are strongly recommended to justify the importance of social media influencers in driving engagement in the paper. Please read and cite the following papers: 

Cheung, M. L., Leung, W. K., Aw, E. C. X., & Koay, K. Y. (2022). “I follow what you post!”: The role of social media influencers’ content characteristics in consumers' online brand-related activities (COBRAs). Journal of Retailing and Consumer Services66, 102940.

Lou, C., Tan, S. S., & Chen, X. (2019). Investigating consumer engagement with influencer-vs. brand-promoted ads: The roles of source and disclosure. Journal of Interactive Advertising19(3), 169-186.

Argyris, Y. A., Wang, Z., Kim, Y., & Yin, Z. (2020). The effects of visual congruence on increasing consumers’ brand engagement: An empirical investigation of influencer marketing on instagram using deep-learning algorithms for automatic image classification. Computers in Human Behavior112, 106443.

The authors are recommended to discuss the behavioural engagement by presenting the importance of COBRAs in the introduction section, and to discuss how social media influencers / social media communities drive engagement by using different marketing elements, such as creative contents, technology, design and informative contents. 

Author Response

(The authors gave the same response as above.)
